# Non-metric Similarity Graphs for Maximum Inner Product Search

**Stanislav Morozov**
Yandex,
Lomonosov Moscow State University
`stanis-morozov@yandex.ru`

**Artem Babenko**
Yandex,
National Research University
Higher School of Economics
`artem.babenko@phystech.edu`

## Abstract

In this paper we address the problem of Maximum Inner Product Search (MIPS) that is currently the computational bottleneck in a large number of machine learning applications. While being similar to the nearest neighbor search (NNS), the MIPS problem was shown to be more challenging, as the inner product is not a proper metric function. We propose to solve the MIPS problem with the usage of similarity graphs, i.e., graphs where each vertex is connected to the vertices that are the most similar in terms of some similarity function. Originally, the framework of similarity graphs was proposed for metric spaces and in this paper we naturally extend it to the non-metric MIPS scenario. We demonstrate that, unlike existing approaches, similarity graphs do not require any data transformation to reduce MIPS to the NNS problem and should be used for the original data. Moreover, we explain why such a reduction is detrimental for similarity graphs. By an extensive comparison to the existing approaches, we show that the proposed method is a game-changer in terms of the runtime/accuracy trade-off for the MIPS problem.

## 1 Introduction

The Maximum Inner Product Search (MIPS) problem has recently received increased attention from different research communities. The machine learning community has been especially active on this subject, as MIPS arises in a number of important machine learning tasks such as efficient Bayesian inference[1, 2], memory networks training[3], dialog agents[4], reinforcement learning[5]. The MIPS problem formulates as follows. Given the large database of vectors $X = \{x_i \in \mathbb{R}^d | i = 1, \ldots, n\}$ and a query vector $q \in \mathbb{R}^d$, we need to find an index $j$ such that

$$\langle x_j, q \rangle \geq \langle x_i, q \rangle = x_i^T q, i \neq j \tag{1}$$

In practice we often need $K > 1$ vectors that provide the largest inner products and the top-$K$ MIPS problem is considered.

For large-scale databases the sequential scan with the $O(nd)$ complexity is not feasible, and the efficient approximate methods are required. The current studies on efficient MIPS can be roughly divided into two groups. The methods from the first group [6, 7, 8], which are probably the more popular in the machine learning community, reduce MIPS to the NNS problem. They typically transform the database and query vectors and then search the neighbors via traditional NNS structures, e.g., LSH[7] or partition trees[6]. The second group includes the methods that filter out the unpromising database vectors based on inner product upper bounds, like the Cauchy-Schwarz inequality[9, 10].

In this work we introduce a new research direction for the MIPS problem. We propose to employ the similarity graphs framework that was recently shown to provide the exceptional performance for the nearest neighbor search. In this framework the database is represented as a graph, where each

vertex corresponds to a database vector. If two vertices $i$ and $j$ are connected by an edge that means that the corresponding database vectors $x_i$ and $x_j$ are close in terms of some metric function. The neighbor search for a query $q$ is performed via graph exploration: on each search step, the query moves from the current vertex to one of adjacent vertex, corresponding to a vector, which is the closest to the query. The search terminates when the query reaches a local minimum. To the best of our knowledge, we are the first who expands similarity graphs on the MIPS territory with non-metric similarity function. We summarize the main contributions of this paper below:

1. We provide the theoretical analysis to justify the use of similarity graphs for inner product similarity function.

2. We demonstrate both theoretically and experimentally that typical MIPS-to-NNS reductions are detrimental for similarity graphs.

3. We introduce a new large-scale dataset for the MIPS algorithms evaluation to facilitate research in this direction. The dataset and the C++ implementation of our method are available online[1].

The rest of the paper is organized as follows: in Section 2, we shortly review the existing MIPS methods and the similarity graphs framework. In Section 3 we advocate the usage of similarity graphs for MIPS and describe the efficient algorithm as well. In addition, we demonstrate that one should not reduce MIPS to NNS when using similarity graphs. In Section 4, we compare the proposed approach to the current state-of-the-art and demonstrate its exceptional advantage over existing methods. Finally, in Section 5 we conclude the paper and summarize the results.

## 2 Related work

Now we describe several methods and ideas from the previous research that are essential for description of our method. Hereafter we denote the database by $X = \{x_i \in \mathbb{R}^d | i = 1, \ldots, n\}$ and a query vector by $q \in \mathbb{R}^d$.

### 2.1 The existing approaches to the MIPS problem

**Reduction to NNS.** The first group of methods[6, 7, 8] reformulates the MIPS problem as a NNS problem. Such reformulation becomes possible via mapping the original data to a higher dimensional space. For example, [7] maps a database vector $x$ to

$$\hat{x} = (x, \sqrt{1 - \|x\|^2})^T \tag{2}$$

and a query vector $q$ is mapped to

$$\hat{q} = (q, 0)^T \tag{3}$$

The transformation from [7] assumes without loss of generality that $\|x\| \leq 1$ for all $x \in X$ and $\|q\| = 1$ for a query vector $q$. After the mapping the transformed vectors $\hat{x}$ and $\hat{q}$ have unit norms and

$$\|\hat{x} - \hat{q}\|^2 = \|\hat{x}\|^2 + \|\hat{q}\|^2 - 2\langle \hat{x}, \hat{q} \rangle = -2\langle x, q \rangle + 2 \tag{4}$$

so the minimization of $\|\hat{x} - \hat{q}\|$ is equivalent to the maximization of $\langle x, q \rangle$. Other MIPS-to-NNS transformations are also possible, as shown in [11] and [12], and the empirical comparison of different transformations was recently performed in [6]. After transforming the original data, the MIPS problem becomes equivalent to metric neighbor search and can be solved with standard NNS techniques, like LSH[7], Randomized Partitioning Tree[6] or clustering[8].

**Upper-bounding.** Another family of methods use inner product upper bounds to construct a small set of promising candidates, which are then checked exhaustively. For example, the LEMP framework[9] filters out the unpromising database vectors based on the Cauchy-Schwarz inequality. Furthermore, [9] proposes an incremental pruning technique that refines the upper bound by computing the partial inner product over the first several dimensions. The FEXIPRO method[10] goes further and performs SVD over the database vectors to make the first dimensions more meaningful. These steps typically improve the upper bounds, and the incremental pruning becomes more efficient. The very recent

Greedy-MIPS method[13] uses another upper bound $\langle q, x \rangle = \sum\limits_{i=1}^{d} q_i x_i \leq d \max_i \{q_i x_i\}$ to construct the candidate set efficiently. The efficiency of upper-bound methods is confirmed by experimental comparison[9, 13] and their source code, available online.

Despite a large number of existing MIPS methods, the problem is far from being solved, especially given rapidly growing databases in nowadays applications. In this work we propose to solve MIPS with the similarity graphs framework that does not fall into either of the two groups above.

## 2.2 NNS via similarity graph exploration

Here we shortly describe the similarity graphs that are currently used for NNS in metric spaces. For a database $X = \{x_i \in \mathbb{R}^d | i = 1, \ldots, n\}$ the similarity (or knn-)graph is a graph where each vertex corresponds to one of the database vectors $x$. The vertices $i$ and $j$ are connected by an edge if $x_j$ belongs to the set of k nearest neigbors of $x_i$, $x_j \in NN_k(x_i)$ in terms of some metric similarity function $s(x, y)$. The usage of knn-graphs for NNS was initially proposed in the seminal work[14]. The approach[14] constructs the database knn-graph and then performs the search by greedy walk on this graph. First, the search process starts from a random vertex and then on each step a query moves from the current vertex to its neighbor, which appears to be the closest to a query. The process terminates when the query reaches a local minimum. The pseudocode of the greedy walk procedure is presented on Algorithm 1.

---

**Algorithm 1** Greedy walk

---

1: **Input:** Similarity Graph $G_s$, similarity function $s(x, y)$, query $q$, entry vertex $v_0$
2: Initialize $v_{curr} = v_0$
3: **repeat**
4:     **for** $v_{adj}$ adjacent to $v_{curr}$ in $G_s$ **do**
5:         **if** $s(v_{adj}, q) < s(v_{curr}, q)$ **then**
6:             $v_{curr} = v_{adj}$
7: **until** $v_{curr}$ changes
8: return $v_{curr}$

---

Since [14] gave rise to research on NNS with similarity graphs, a plethora of methods, which elaborate the idea, were proposed. The current state-of-the-art graph-based NNS implementations[15, 16, 17] develop additional heuristics that increase the efficiency of both graph construction and search process. Here we describe in detail the recent Navigable Small World (NSW) approach[15], as it is shown to provide the state-of-the-art for NNS[18] and its code is available online. Our approach for MIPS will be based on the NSW algorithm, although the other graph-based NNS methods[16, 17] could also be used.

---

**Algorithm 2** NSW graph construction

---

1: **Input:** Database $X$, similarity function $s(x, y)$, maximum vertex degree $M$
2: Initialize graph $G_s = \emptyset$
3: **for** $x$ in $X$ **do**
4:     $S = \{M$ vertices from $G_s$, s.t. the corresponding vectors $y$ give the largest values of $s(x, y)\}$
5:     Add $x$ to the graph $G_s$ and connect it by the directed edges with vertices in $S$
6: return $G_s$

---

The key to the practical success of NSW lies in the efficiency of both knn-graph construction and neighbor search. NSW constructs the knn-graph by adding vertices in the graph sequentially one by one. On each step NSW adds the next vertex $v$, corresponding to a database vector $x$ to the current graph. $v$ is connected by directed edges to $M$ vertices, corresponding to the closest database vectors that are already added to the graph. The construction algorithm is presented in Algorithm 2. The primary parameter of the NSW is the maximum vertex degree $M$, which determines the balance between the search efficiency and the probability that search stops in the suboptimal local minima. When searching via Greedy walk, NSW maintains a priority queue of a size $L$ with the knn-graph vertices, which neighbors should be visited by the search process. With $L=1$ the search in NSW is equivalent to Algorithm 1, while with $L > 1$ it can be considered as a variant of Beam Search[19],

which makes the search process less greedy. In practice, varying $L$ allows to balance between the runtime and search accuracy in NSW.

**Prior work on non-metric similarity search on graphs.** After the publication, we became aware of a body of previous work that explored the use of proximity graphs with general non-metric similarity functions[20, 21, 22, 23, 24]. In these works, the MIPS problem is investigated as a special case and the effectiveness of proximity graph based methods to the MIPS problem has been confirmed.

# 3   Similarity graphs for MIPS

Now we extend the similarity graphs framework to applications with a non-metric similarity function $s(x, y)$. Assume that we have a database $X = \{x_i \in \mathbb{R}^d | i = 1, \ldots, n\}$ and aim to solve the problem

$$\arg\max_{x_i \in X} s(q, x_i), \ q \in \mathbb{R}^d \tag{5}$$

## 3.1   Exact solution

First, let us construct a graph $G_s$ such that the greedy walk procedure (Algorithm 1), provides the exact answer to the problem (5). [14] has shown that for Euclidean distance $s(x, y) = -\|x - y\|$, the minimal $G_s$ with this property is the Delaunay graph of $X$. Now we generalize this result for a broader range of similarity functions.

**Definition.** The $s$-Voronoi cell $R_k$, associated with the element $x_k \in X$, is a set

$$R_k = \{x \in \mathbb{R}^d | s(x, x_k) > s(x, x_j) \ \forall j \neq k\} \tag{6}$$

The diagrams of $s$-Voronoi cells for $s(x, y) = -\|x - y\|$ and $s(x, y) = \langle x, y \rangle$ are shown on Figure 1.

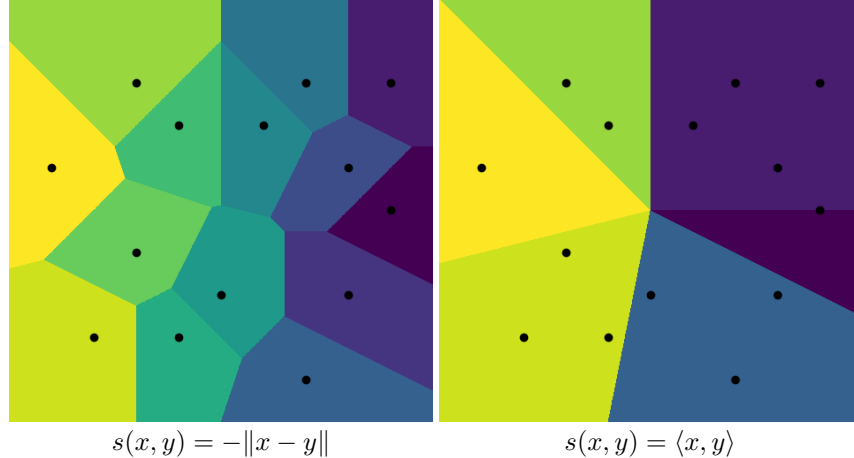

$$s(x, y) = -\|x - y\| \qquad\qquad s(x, y) = \langle x, y \rangle$$

Figure 1: $s$-Voronoi diagram examples on the plane

Note, that in the case of inner product $s$-Voronoi cells for some points are empty. It implies that these points can not be answers in MIPS.

Now we can define a $s$-Delaunay graph for a similarity function $s(x, y)$.

**Definition.** The $s$-Delaunay graph for the database $X$ and the similarity function $s(x, y)$ is a graph $G_s(V, E)$ where the set of vertices $V$ corresponds to the set $X$ and two vertices $i$ and $j$ are connected by an edge if the correspoding $s$-Voronoi cells $R_i$ and $R_j$ are adjacent in $\mathbb{R}^d$.

**Theorem 1.** *Suppose that the similarity function $s(x, y)$ is such that for every finite database $X$ the corresponding $s$-Voronoi cells are path-connected sets. Then the greedy walk (Algorithm 1) stops at the exact solution for problem (5) if the similarity graph $G_s$ contains the $s$-Delaunay graph as a subgraph.*

*Proof.* Assume that the greedy walk with a query $q$ stops at the point $x$ i.e. $s(x, q) > s(y, q)$ for all $y \in N(x)$, where $N(x)$ is a set of vertices that are adjacent to $x$. Suppose that there is a point $z \notin N(x)$ such that $s(z, q) > s(x, q)$. It means that the point $q$ does not belong to the $s$-Voronoi cell $R_x$ corresponding to the point $x$. Note, that if we remove all the points from $G_s$ except $x \cup N(x)$, a set of points covered by $R_x$ does not change as all adjacent $s$-Voronoi regions correspond to vertices from $N(x)$ and they are not removed. Hence, the query $q$ still does not belong to $R_x$. Since the $s$-Voronoi cells cover the whole space, the point $q$ belongs to some $R_{x'}, x' \in N(x)$. This means that $s(x', q) > s(x, q)$. This contradiction proves the theorem. $\qquad \square$

Now we show that $s(x, y) = \langle x, y \rangle$ satisfies the assumptions of the Theorem.

**Lemma 1.** *Suppose $X$ is a finite database and the similarity function $s(x, y)$ is linear, then the $s$-Voronoi cells are convex.*

*Proof.* Consider a $s$-Voronoi cell $R_x$, corresponding to a point $x \in X$. Let us take two arbitrary vectors $u$ and $v$ from the $s$-Voronoi cell $R_x$. It means that

$$s(x, u) > s(w, u) \; \forall w \in X \setminus \{x\} \tag{7}$$

$$s(x, v) > s(w, v) \; \forall w \in X \setminus \{x\} \tag{8}$$

Hence, due to linearity

$$s(x, tu + (1 - t)v) > s(w, tu + (1 - t)v), \; t \in [0, 1] \tag{9}$$

Therefore, vector $tu + (1 - t)v \in R_x$ for every $t \in [0, 1]$. $\qquad \square$

**Corollary 1.** *If the graph $G(V, E)$ contains the $s$-Delaunay graph for the similarity function $s(x, y) = \langle x, y \rangle$ then greedy walk always gives the exact true answer for MIPS.*

*Proof.* Due to Lemma 1 all $s$-Voronoi cells for $s(x, y) = \langle x, y \rangle$ are convex, therefore, path-connected. $\qquad \square$

### 3.2 $s$-Delaunay graph approximation for MIPS

In practice, the computation and usage of the exact $s$-Delaunay graph in high-dimensional spaces are infeasible due to the exponentially growing number of edges[25]. Instead, we approximate the $s$-Delaunay graph as was previously proposed for Euclidean distance case in [16, 17, 15]. In particular, we adopt the approximation proposed in [15] by simply extending Algorithm 2 to inner product similarity function $s(x, y) = \langle x, y \rangle$. As in [15] we also restrict the vertex degree to a constant $M$, which determines the $s$-Delaunay graph approximation quality. We refer to the proposed MIPS method as *ip-NSW*. The search process in ip-NSW remains the same as in [15] except that the inner product similarity function guides the similarity graph exploration.

Let us provide some intuition behind the proposed $s$-Delaunay graph approximation. In fact, each vertex $x$ is connected to $M$ vertices that provide the highest inner product values $\langle x, \cdot \rangle$. The heuristic geometrical argument in favor of such approximation is that for $s(x, y) = \langle x, y \rangle$ $s$-Voronoi cells are polyhedral angles, and the «direction vectors» of adjacent $s$-Voronoi cells are likely to have large inner product values. While missing the strict mathematical justification, the proposed approach provides the brilliant performance, as confirmed in the experimental section.

### 3.3 Similarity graphs after reduction to NNS

The natural question is: Why should we develop an additional theory for non-metric similarity graphs? Maybe, one should just reduce the MIPS problem to NNS[6, 7, 8] and apply the state-of-the-art graph implementation for Euclidean similarity. In fact, such a solution is detrimental for runtime-accuracy trade-off, as will be demonstrated in the experimental section. In this section, we provide the intuitive explanation of the inferior performance using the example of transformation form [7]:

$$\hat{x} = (x, \sqrt{1 - \|x\|^2})^T; \quad \hat{q} = (q, 0)^T = (q, \sqrt{1 - \|q\|^2})^T \tag{10}$$

assuming that $\|x\| \leq 1$ for all $x \in X$ and $\|q\| = 1$. Other transformations could be considered in the similar way. Now we construct the Euclidean similarity graph for the transformed database

$\hat{X} = \{(x, \sqrt{1 - \|x\|^2})^T | x \in X\}$ via Algorithm 2. In terms of the original database $X$, the Euclidean distance between the transformed elements equals

$$\|\hat{x} - \hat{y}\|^2 = -2\langle x, y \rangle + 2 - 2\sqrt{1 - \|x\|^2}\sqrt{1 - \|y\|^2} \qquad (11)$$

Note, that the Euclidean similarity graph, constructed for the transformed database $\hat{X}$, is equivalent to a graph, constructed for the original $X$ with the similarity function $s(x, y) = \langle x, y \rangle + \sqrt{1 - \|x\|^2}\sqrt{1 - \|y\|^2}$ or equivalently

$$s(x, y) = \|x\|\|y\| \cos \alpha + \sqrt{1 - \|x\|^2}\sqrt{1 - \|y\|^2}, \qquad (12)$$

where $\alpha$ is the angle between $x$ and $y$. The first term in this sum encourages large norms, while the second term penalizes large norms. In high-dimensional spaces the typical values of $\cos \alpha$ tend to be small even for close vectors, which results in the dominance of the second term. Thus, when a new vertex is added to a graph, it prefers to be connected to the vertices, corresponding to vectors with smaller norms. Thus, the edges in the Euclidean graph, constructed for the transformed data, typically lead in the direction of norm decreasing, which is counterproductive to MIPS, which prefers the vectors of larger norms. On the other hand, the non-metric similarity graph, constructed with $s(x, y) = \langle x, y \rangle$, is more probable to contain edges, directed towards increasing of norms. To verify this explanation, we measure the rate of edges that lead to vectors of larger norms for ip-NSW and the Euclidean NSW on the transformed data. The numbers for three datasets, presented in Table 2, fully confirm our intuition.

## 4 Experiments

In this section we present the experimental evaluation of non-metric similarity graphs for the top-$K$ MIPS problem. All the experiments were performed on Intel Xeon E5-2650 machine in a single thread mode. For evaluation, we used the commonly used *Recall* measure that is defined as a rate of successfully found neighbors, averaged over a set of queries. We performed the experiments with $K = 1$ and $K = 10$.

**Datasets.** We summarize the information on the benchmark datasets in Table 1. The *Netflix*, *MovieLens* and *Yahoo!Music* datasets are the established benchmarks for the MIPS problem. *Music-100* is a new dataset that we introduce to the community[2]. This dataset was obtained by IALS-factorization[26] of the user-item ranking matrix, with dimensionality 100. The matrix contains the ratings from 3,897,789 users on one million popular songs from proprietary music recommendation service. To the best of our knowledge, there is no publicly available dataset of such a large scale and high dimensionality. *Normal-64* dataset was generated as a sample from a standard normal distribution with the dimension 64. For all the datasets, the groundtruth neighbors were computed by sequential scan. The recall values were averaged over 10,000 randomly sampled queries.

Table 1: The datasets used in the evaluation.

| DATASET | $|X|$ | $|Q|$ | DIM |
|---|---|---|---|
| NETFLIX | 17,770 | 480,189 | 200 |
| MOVIELENS | 33,670 | 247,753 | 150 |
| YAHOO! MUSIC | 624,961 | 1,000,990 | 50 |
| MUSIC-100 | 1,000,000 | 3,897,789 | 100 |
| NORMAL-64 | 1,048,576 | 20,000 | 64 |

### 4.1 Non-metric graphs or reduction to NNS?

Here we experimentally investigate the optimal way to use the similarity graphs for the MIPS problem. We argue that the straightforward solution by reduction to NNS and then using the standard Euclidean similarity graph is suboptimal. To confirm this claim, we compare the performance of the non-metric similarity graph (denoted by ip-NSW) to the performance of Euclidean similarity graph combined with transformation from[7] (denoted by *NSW+reduction*). The runtime-accuracy

plots on three datasets are presented on Figure 2. The plots confirm the advantage of non-metric similarity graphs, especially in the high recall regime. For instance, ip-NSW reaches the recall level 0.9 five times faster on Music-100. We believe that the reason for the inferior performance of the NSW+reduction approach is the edge distribution bias, described in Section 3.3. Overall, we conclude that similarity graphs do not require any MIPS-to-NNS transformation that makes them favorable over other similarity search frameworks. In the subsequent experiments, we evaluate only the ip-NSW approach as our main contribution.

Table 2: The rate of similarity graph edges that lead to vector of larger norms for ip-NSW and NSW+reduction. This rate is much higher in the non-metric similarity graph in ip-NSW, which results in higher MIPS performance.

| DATASET | NSW+REDUCTION | IP-NSW |
|---|---|---|
| MUSIC-100 | 0.349335 | 0.75347 |
| YAHOO! MUSIC | 0.398541 | 0.92353 |
| NORMAL-64 | 0.362722 | 0.703605 |

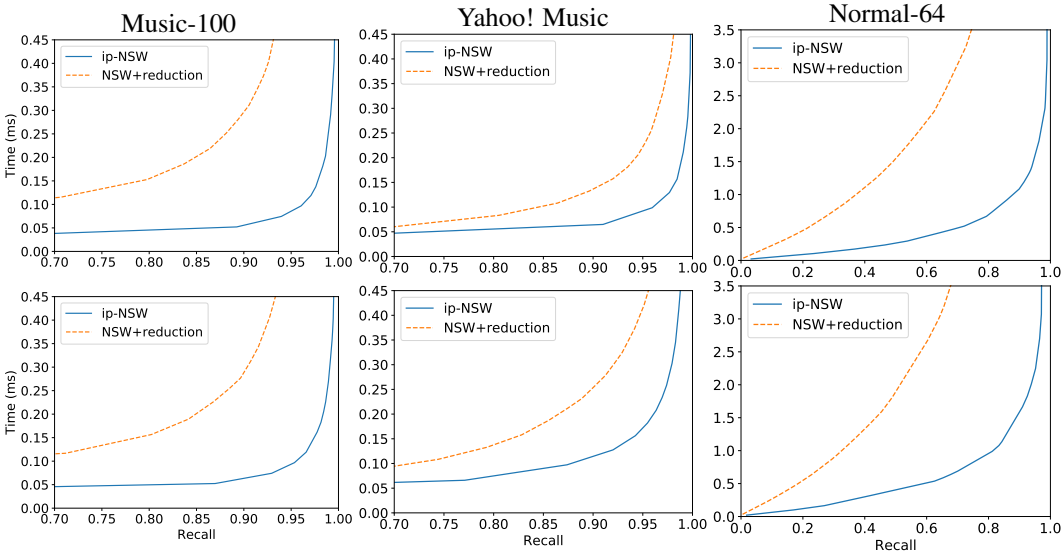

Figure 2: The performance of non-metric ip-NSW and the Euclidean NSW for transformed data on three million-scala datasets. The combination of metric similarity graphs with MIPS-to-NNS reduction results in inferior performance.

### 4.2 Comparison to the state-of-the-art

As our main experiment, we extensively compare the proposed ip-NSW method to the existing approaches. We compared the following algorithms:

**Naive-MKL** The sequential scan implementation that uses the Intel MKL library[3] for efficient vector-matrix multiplication.

**LSH+reduction[7]** We used the implementation available in [13]. We tuned the parameter $B$ in a range $\{20, 40, 80, 160\}$ and the parameter $R$ in a range $\{5, 8, 11, 14, 17, 20\}$.

**Clustering+reduction[8]** We used our own reimplementation and use $\sqrt{N}$ clusters of size $\sqrt{N}$, where $N$ is the size of the database. When searching, the number of considered clusters was varied from 1 to 50.

**FEXIPRO[10]** We used the author's implementation of the FEXIPRO framework[4] with the algorithm FEXIPRO-SIR and the parameters $scalingValue = 127$ and $SIGMA = 0.8$ since it was recommended in [10] as the best combination. Note, that FEXIPRO is an exact method.

**LEMP[9]** We used the author's implementation of the LEMP framework[5] with the algorithm LEMP-HYB-REL. We varied parameters $R$ from $0.1$ to $0.9$ with step $0.1$ and $\varepsilon$ from $0.2$ to $0.6$ with step $0.05$ to achieve the runtime-accuracy plots.

**Greedy-MIPS[13]** We used the author's implementation[6] with a $budget$ parameter tuned for the each dataset.

**ip-NSW** is the proposed algorithm based on the non-metric similarity graph, described in Section 3.2.

While Netflix and MovieLens are the established datasets in previous works, we do not consider them as interesting benchmarks these days. They both contain only several thousand vectors and the exact Naive-MKL is efficient enough on them. E.g. Naive-MKL works only $0.56$ ms on Netflix and $1.42$ ms on MovieLens, which is fast enough for most of applications. Thus, we perform the extensive comparison on three million-scale datasets only. The Figure 3 presents the runtime-accuracy plots for the compared approaches. The timings for Naive-MKL and FEXIPRO are presented under the corresponding plots. Overall, the proposed ip-NSW method outperforms the existing approaches by a substantial margin. For example, ip-NSW reaches $0.9$ recall level ten times faster that the fastest baseline. Note, that the for top-10 MIPS the advantage of ip-NSW is even more impressive on all datasets. To justify that the speedup improvements are due to the proposed algorithm and not because of implementation differences (such as libraries, cache locallity, register level optimizations and so on) we also compare number of inner products needed to achieve certain recall levels for different methods. The plots for three datasets and top-10 MIPS are presented on Figure 4.

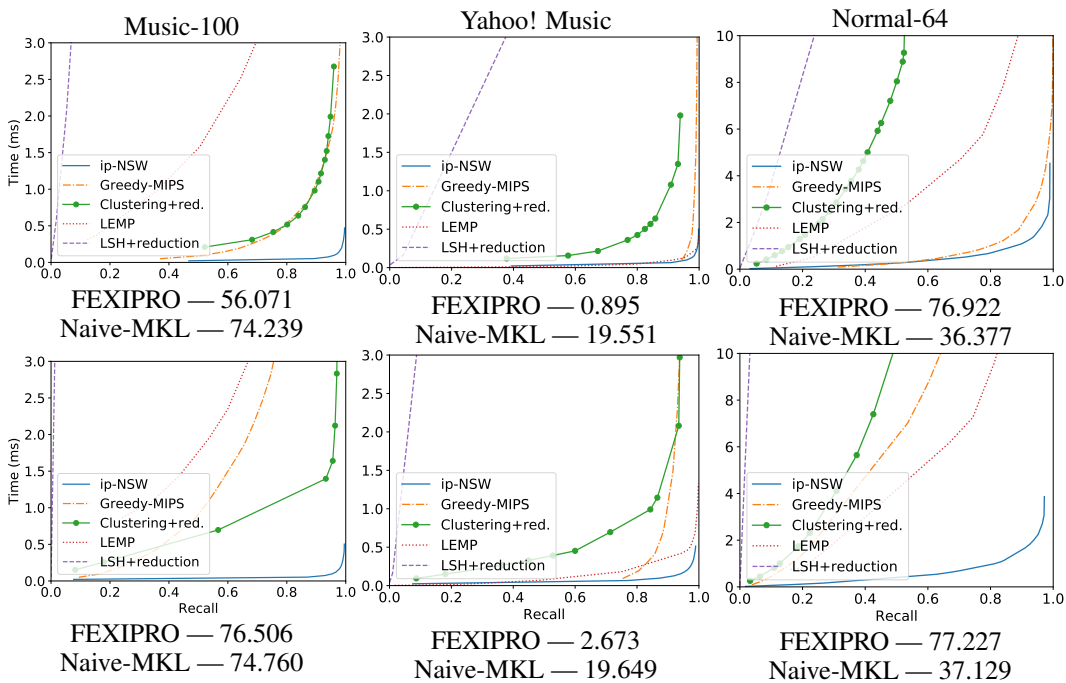

Figure 3: The runtime-recall plots on three datasets for top-1 MIPS (top) and top-10 MIPS (bottom). The timings for the exact FEXIPRO and Naive-MKL methods are presented under the corresponding plots.

**Additional memory consumption.** The performance advantage of the similarity graphs comes at a price of additional memory to maintain the graph structure. In our experiments we use $M = 32$ edges per vertex, which results in $32 \times n \times sizeof(int)$ bytes for edge lists. Note, that the size of the database equals $d \times n \times sizeof(float)$ bytes, hence for high-dimensional datasets $d \gg 32$ the additional memory consumption is negligible.

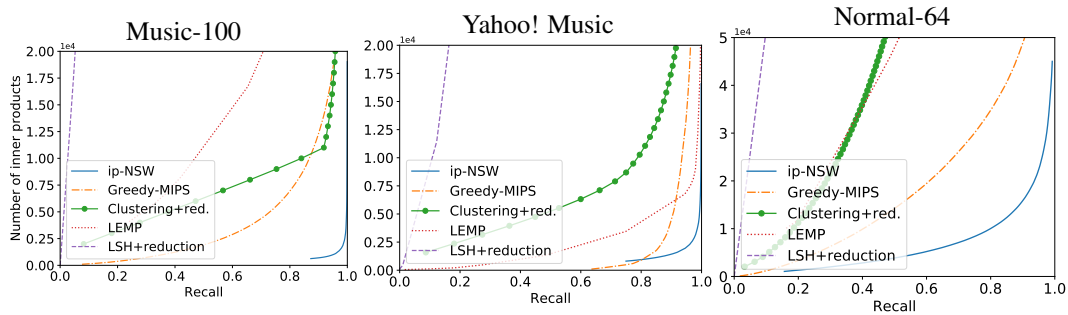

Figure 4: The number of inner products computations needed to achieve certain recall levels on three datasets for top-10 MIPS.

## 5 Conclusion

In this work, we have proposed and evaluated a new framework for inner product similarity search. We extend the framework of similarity graphs to the non-metric similarity search problems and demonstrate that the practically important case of inner product could be perfectly solved by these graphs. We also investigate the optimal way to use this framework for MIPS and demonstrate that the popular MIPS-to-NNS reductions are harmful to similarity graphs. The optimized implementation of the proposed method will be available upon publication to support the further research in this direction.

## Footnotes

[1]https://github.com/stanis-morozov/ip-nsw

[2]`https://github.com/stanis-morozov/ip-nsw`

[3]https://software.intel.com/mkl

[4]https://github.com/stanford-futuredata/FEXIPRO-orig

[5]`https://github.com/uma-pi1/LEMP`

[6]`https://github.com/rofuyu/exp-gmips-nips17`

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
