[Reviews · NeurIPS 2018]

Reviewer 1



Summary: The authors present a new technique for MIPS that is closely related to (if not a direct application) an existing technique for NNS. The authors provide some theoretical backing for the algorithm, though not sufficient to prove the performance for the parameter settings that would be used in practice (which is typical for the field). Finally, they show experiments on two real-world music ratings datasets and a synthetic dataset. Quality: The technique seems natural, relatively simple (a good thing), and yields strong performance. I’m not familiar enough with the MIPS literature to know if the list of baselines is complete. It is somewhat disappointing that the datasets were not very comprehensive. Only two real world datasets were used which are similar to each other (both music ratings datasets). Clarity: The paper is easy to read and ordered in an intuitive way. Enough technical detail is included to make the paper concrete, but not too much. Originality: While the idea of a nearest neighbor graph may be novel for the MIPS problem, it already exists for the NNS problem for metric spaces. The originality is probably the weakest part of this paper. Significance: The MIPS problem is a fundamental problem in machine learning and other fields. The experiments lead me to believe that this technique provides major speed-ups on several datasets, both with real-world structure and drawn with uniform directions. Small Comment: Line 5 of Algorithm 1 should it be s instead of d?

Reviewer 2



The authors identify the issue in solving MIPS by reducing it to NNS and propose the use of non-metric similarity graphs to solve MIPS in its native form. While the idea and the empirical results are extremely promising, there are a couple of issues that temper my enthusiasm for this manuscript significantly. Major: - While the theoretical results claim the correctness of the Greedy Walk algorithm (Alg 1) with s-Delaunay graphs, it is hard for me to understand (concretely or even intuitively) how the approximation in the graph construction (so that it is no longer a s-Delaunay graph) affects the accuracy of Alg 1 for MIPS. With existing methods such as LSH/RP-trees+reduction or Greedy-MIPS, the algorithm will find the exact answer(or within some approximation) eventually or drive the probability of not finding the exact answer to really low values (with RP-trees providing stronger guarantees than LSH). However, with Alg 1, there is no guarantee on the approximation of the obtained answer. - Empirical results in terms of actual runtimes are a great way of evaluating the proposed solution. However, it makes it hard for readers to judge whether the improvements of the proposed solution because of the novel/proposed algorithm (which is discussed in detail in the manuscript) or because of implementation details such as programming languages/libraries used, any kind of cache locality/multithreading/register level optimizations used, etc (which are not discussed/elaborated/focused on in the manuscript and hence unknown to the reader). And these implementation details can be quite significant when we are discussing runtimes in the order of milliseconds (unless these numbers are generated by repeating the runs multiple times and ensuring there are no competing programs on the evaluation machine). An accompanying empirical comparison using number of vector operations (for example, dot products) as a surrogate for time would provide a fairer comparison of the algorithms being evaluated, and make the gains from the proposed algorithm more clear. Minor: - How are the recall vs. time curves generated for the proposed graph based solution in Figures 2,3? What parameters of the algorithm are being varied to span the complete speed-accuracy tradeoff? This is not clear to me from the description of Alg 1 and Section 4. - As per lines 41-42, the authors claim to provide "theoretical analysis to justify the use of similarity graphs". However, I couldn't find any concrete theoretical result supporting this claim (unless I am missing something). - How does the value of the parameters M (graph vertex degree) and L (the size of the priority queue maintained during the search) affect the performance of the proposed MIPS solution? Is the proposed method sensitive to these parameters? How dependent are these parameters on the dataset? ================ Post author-response edit: I want to thank the authors for their thoughtful responses; I acknowledge that I have read them. The authors have addressed two of my major concerns and I am thoroughly satisfied with one. However, the other concern (re: approximation-accuracy discussion), I appreciate the explanation and intuition -- it definitely helps me understand things better, but it still lacks certain amount of rigour, and the fact that there is no recovery from a local minima (or alternately, no guarantees on the local minima relative to the global minima) still leaves me somewhat unsatisfied. However, my score is still fairly positive so I will leave it at that.

Reviewer 3



This paper addresses the Maximum Inner Product Search (MIPS) problem by using the popular Approximate Nearest Neighbor Search (ANN Search) technique: Navigable Small World (NSW) graphs. The solution is just replacing the metric distance functions, such as Cosine and L2, by the non-metric function Inner Product. And they claim that this method, ip-NSW, is better than NSW+reduction, which is one of popular solutions for MIPS. They provide theoretical analysis for that. Experimental results show that ip-NSW works very well on MIPS. Weak points or concerns: (1) It was claimed that this paper is the first to expand similarity graphs on MIPS. But as far as I know there are at least two open source ANN search platforms which implemented Inner Product Search by Hierarchical Navigable Small World (HNSW) graphs (links as below). So I think the novelty of this paper is limited. - https://github.com/nmslib/nmslib - https://github.com/nmslib/hnsw (2) The theoretical analysis sounds interesting because the theory analysis for searching in non-metric space is lack in the area. But I should say that the NSW method just builds approximate Delaunay graphs. There is no guarantee, in which level, the property of Delaunay graph holds on the built NSW graph, especially for non-metric distances. The theoretical analysis is only for exact Delaunay graphs but not for approximate ones. So this part is weak also.